# The Impact of ZIP8 Disease-Associated Variants G38R, C113S, G204C, and S335T on Selenium and Cadmium Accumulations: The First Characterization

**DOI:** 10.3390/ijms222111399

**Published:** 2021-10-22

**Authors:** Zhan-Ling Liang, Heng Wee Tan, Jia-Yi Wu, Xu-Li Chen, Xiu-Yun Wang, Yan-Ming Xu, Andy T. Y. Lau

**Affiliations:** Laboratory of Cancer Biology and Epigenetics, Department of Cell Biology and Genetics, Shantou University Medical College, Shantou 515041, China; 16zlliang@stu.edu.cn (Z.-L.L.); hwtan@stu.edu.cn (H.W.T.); jiayiwu1@163.com (J.-Y.W.); 19xlchen@stu.edu.cn (X.-L.C.); 19xywang@stu.edu.cn (X.-Y.W.)

**Keywords:** cadmium cytotoxicity, cancer therapy, cisplatin, ICP-MS, nonsynonymous mutation, selenium homeostasis, selenoproteins, ZIP8

## Abstract

The metal cation symporter ZIP8 (SLC39A8) is a transmembrane protein that imports the essential micronutrients iron, manganese, and zinc, as well as heavy toxic metal cadmium (Cd). It has been recently suggested that selenium (Se), another essential micronutrient that has long been known for its role in human health and cancer risk, may also be transported by the ZIP8 protein. Several mutations in the ZIP8 gene are associated with the aberrant ion homeostasis of cells and can lead to human diseases. However, the intricate relationships between ZIP8 mutations, cellular Se homeostasis, and human diseases (including cancers and illnesses associated with Cd exposure) have not been explored. To further verify if ZIP8 is involved in cellular Se transportation, we first knockout (KO) the endogenous expression of ZIP8 in the HeLa cells using the CRISPR/Cas9 system. The elimination of ZIP8 expression was examined by PCR, DNA sequencing, immunoblot, and immunofluorescence analyses. Inductively coupled plasma mass spectrometry indicated that reduced uptake of Se, along with other micronutrients and Cd, was observed in the ZIP8-KO cells. In contrast, when ZIP8 was overexpressed, increased Se uptake could be detected in the ZIP8-overexpressing cells. Additionally, we found that ZIP8 with disease-associated single-point mutations G38R, G204C, and S335T, but not C113S, showed reduced Se transport ability. We then evaluated the potential of Se on Cd cytotoxicity prevention and therapy of cancers. Results indicated that Se could suppress Cd-induced cytotoxicity via decreasing the intracellular Cd transported by ZIP8, and Se exhibited excellent anticancer activity against not all but only selected cancer cell lines, under restricted experimental conditions. Moreover, clinical-based bioinformatic analyses revealed that up-regulated ZIP8 gene expression was common across multiple cancer types, and selenoproteins that were significantly co-expressed with ZIP8 in these cancers had been identified. Taken together, this study concludes that ZIP8 is an important protein in modulating cellular Se levels and provides insights into the roles of ZIP8 and Se in disease prevention and therapy.

## 1. Introduction

Selenium (Se) is an essential micronutrient critical for maintaining normal cellular function in human and animal cells [1]. It is an integral component of selenoproteins which are involved in a wide range of cellular physiological processes, including but not limited to antioxidant defense, inflammatory response, immune regulation, and maintenance of cardiovascular and reproductive system [2,3,4]. A balanced Se level is vital to human health as it has been broadly recognized that Se deficiency is one of the causative factors for many human diseases (e.g., heart failure, male infertility, neurodegenerative disease, Keshan disease, and Kashin-Beck disease) [5,6,7] while excessive Se intake is associated with acute or chronic Se poisoning, the selenosis [8].

Se also plays an important role in cancer prevention and therapy [9]. Although some conflicting results have been obtained over the years, epidemiologic studies generally agree that Se deficiency is significantly associated with greater cancer risk, especially in gastrointestinal and prostate cancer [10]. Se and Se compounds have displayed promising anticancer activity against some cancer cell types [11]. However, none of these compounds have yet been clinically recognized as anticancer agents mostly due to inconsistent outcomes within and between clinical trials and laboratory studies [12,13,14]. In addition, Se compounds may be useful in the field of chemoprevention as it has been reported that Se could potentiate the efficacy of some chemotherapeutic drugs, for example, the first-line chemotherapeutic drug cisplatin [15].

Although the effects of Se on human health have received attraction over the past few decades, how Se is being transported into the human cells remains largely unknown. Previous studies have revealed that Se in the body is usually delivered from the liver to other organs or tissues by plasma transporter selenoprotein P (SELENOP/SelP) via interaction with transmembrane receptors such as lipoproteins (e.g., LRP1, LRP2, and LRP8) [16]. Recently, McDermott et al. suggested that another transmembrane protein, metal cation symporter ZIP8, was able to transport Se (in the form of selenite). Furthermore, it was shown that the transport of Se via ZIP8 required zinc (Zn) ion and bicarbonate as co-substrates [17].

ZIP8 is a member of the solute carrier gene family (encoded by the gene SLC39A8) that facilitates the cellular uptake of several essential divalent metals such as iron (Fe), manganese (Mn), and Zn [18,19,20], and it is also responsible for the uptake of toxic heavy metal cadmium (Cd) [19]. Researchers have identified several mutations in the ZIP8 gene that can cause aberrant ion homeostasis of cells and lead to human diseases. For example, mutations G38R (c. 112G>C), G204C (c. 610G>T), and S335T (c. 1004G>C) in the ZIP8 gene are known to be associated with type II congenital disorder of glycosylation, which is characterized by intellectual disability, profound psychomotor retardation, hypotonia, strabismus, loss of hearing, and short stature [21,22]. Leigh syndrome, a rare inherited neurodegenerative disease, is caused by C113S (c. 338G>C) mutation in the ZIP8 [23]. However, the intricate relationships between ZIP8 mutations, cellular Se homeostasis, and human diseases (including cancers and diseases associated with Cd exposure) have not been previously explored.

In the current study, we successfully established a ZIP8-knockout (KO) human cell model using the CRISPR/Cas9 system and confirmed that human ZIP8 is involved in the intracellular transportation of Se. ZIP8-KO cells also showed a decreased ability to transport Mn, Zn, and Cd. We then examined the effects of four selected disease-associated ZIP8 single-point mutations (G38R, C113S, G204C, and S335T) on intracellular Se uptake and assessed the relationship between Se and Cd cytotoxicity in these ZIP8-variant and ZIP8-KO cells. Furthermore, the potential anticancer and synergistic effect of Se combined with cisplatin was determined. Lastly, clinical datasets from TCGA database [24] were analyzed for gene expressions of ZIP8 and 25 genes coding for selenocysteine-containing proteins in multiple cancer types. Overall, findings from this study suggest that ZIP8 is an important protein in modulating cellular Se levels that may have implications for disease prevention and therapy.

## 2. Results

### 2.1. Generation and Verification of a ZIP8 Gene Knockout Human Cell Model

To investigate the relationship between ZIP8 and Se transport, we used the CRISPR/Cas9 system to KO the ZIP8 gene in human cervical cancer HeLa cells. Specifically, we designed two sgRNAs targeting the genomic region of ZIP8 among exon 1 and intron 1, and gene sequence analysis indicated that 845 bp of genomic DNA was successfully deleted in that region (Figure 1A). PCR and immunoblot analysis were used to detect the gene and protein status of ZIP8 in the ZIP8-KO HeLa cell model. Gel electrophoresis assay showed that the HeLa parental cells with wildtype (WT) ZIP8 contained a full-length ZIP8 PCR product (1207 bp) whereas the ZIP8-KO cells generated a ZIP8 PCR product with only 362 bp (Figure 1B, Appendix A). On the other hand, immunoblot analysis showed that the ZIP8 protein expression was dramatically decreased in the ZIP8-KO cells when compared with the HeLa parental cells (Figure 1C). Furthermore, we performed an immunofluorescence assay to visualize the expression and localization of ZIP8 in the ZIP8-WT and ZIP8-KO cells. Microscopic results indicated that ZIP8 in the ZIP8-KO cells had substantially weaker overall expression and did not co-localize with the membrane protein marker DMT1 (Figure 1D). The above results showed that we have successfully generated a ZIP8-KO human cell model.

### 2.2. Involvement of Human ZIP8 in Intracellular Mn, Zn, Cd, and Se Uptakes

ZIP8 is a well-known transporter for certain divalent metal ions, including essential micronutrients Zn and Mn and toxic heavy metal Cd. Thus, we hypothesized that the transport capability of these metals should, to some extent, be affected in the ZIP8-KO cells. To test this hypothesis, we treated the cells with Mn, Zn, or Cd and used inductively coupled plasma mass spectrometry (ICP-MS) to quantify the intracellular uptake of these metals (Figure 2A–C). We found that when compared with HeLa parental cells, the levels of Mn uptake were significantly decreased in the ZIP8-KO cells following the treatment with 100 μM MnCl_2_ at two different time points (Figure 2A). Although the levels of Zn detected in ZIP8-KO cells appeared to be slightly lower than the HeLa parental cells when treated with or without ZnCl_2_, the differences were not statistically significant (Figure 2B). Also, the level of Cd was reduced in the ZIP8-KO cells treated with 1 μM CdCl_2_ for 24 h (Figure 2C). We then used naphthol blue-black (NBB) staining assay to check whether the ZIP8-KO cells, which showed reduced Cd uptake, could survive better against Cd cytotoxicity. As expected, when ZIP8-KO and HeLa parental cells were treated with various concentrations of CdCl_2_ for 12, 24, or 36 h (Figure 2D), greater cell viability was generally observed in ZIP8-KO cells in comparison to HeLa parental cells. These findings strongly suggest that our ZIP8-KO cell line is a reliable cell model to study the biological functions of human ZIP8 further. Also, it is worth noting that a gene sequencing assay was performed to ensure that the ZIP8 gene in HeLa parental cells used in this study has no mutation (data not shown).

The relationship between ZIP8 and Se transport in human cells remains largely unknown. Here, ZIP8-KO cells were used to investigate the effects of ZIP8 on regulating selenite homeostasis. Briefly, HeLa parental and ZIP8-KO cells were exposed to either 200 μM Na_2_SeO_3_ for 10 min or 4 μM Na_2_SeO_3_ for 12 h to mimic acute Se exposure or moderate Se exposure, respectively. The 4 μM Na_2_SeO_3_ treatment was selected to represent “moderate Se exposure” because no cytotoxicity was observed at this concentration, at least for 12 h (Appendix A). Results from the ICP-MS indicated that although Se level appeared to be reduced in the ZIP8-KO cells after supplement with 200 μM Na_2_SeO_3_ for 10 min or 4 μM Na_2_SeO_3_ for 12 h, but only the Se treatment from the latter showed statistical significance (Figure 2E,F). Conversely, the intracellular Se content was enhanced in the ZIP8-overexpressed ZIP8-KO cells (transient-transfected with pcDNA3.1-ZIP8-WT) when compared to the ZIP8-KO control cells expressing pcDNA3.1 empty vector upon Se treatments (Figure 2G). Overall, these results indicated that human ZIP8 is involved in the intracellular uptake of Se.

### 2.3. The Effects of Disease-Associated ZIP8 Single-Point Mutations on Cellular Se Uptake Ability

Studies have shown that single-point mutations, including SNPs, on a single gene can cause abnormal functions of the protein and lead to human diseases [25,26,27]. Several mutations in the ZIP8 have been implicated in the occurrence of diseases related to the dysregulation of ion homeostasis [23]. Here, we selected four disease-associated ZIP8 mutations (G38R, C113S, G204C, and S335T; Table 1) and tested their Se uptake abilities. The selected ZIP8 single-point mutations are illustrated in Figure 3A. ICP-MS was performed to detect the intracellular level of Se in ZIP8-KO cells transiently transfected with pcDNA3.1 empty vector (as ZIP8-negative control) or pcDNA3.1 vector encoding ZIP8-WT or ZIP8-mutant upon acute (Figure 3B) or moderate (Figure 3C) Se exposure. It was demonstrated that with the exception of C113S, the Se transport abilities in all the cells with ZIP8 single-point mutations (G38R, G204C, and S335T) were remarkably suppressed when compared with the ZIP8-WT cells (Figure 3B,C). Particularly, the ZIP8 proteins with G38R, G204C, and S335T mutations appeared to have lost their ability to transport Se during acute Se exposure because the Se concentrations in these mutants were similar to the ZIP8-negative control cells (Figure 3B). However, when these cells were exposed to a moderate concentration of Se (e.g., 4 μM for 12 h), ZIP8-G38R and ZIP8-G204C mutants showed a significant increase in Se uptake compared with the ZIP8-negative control cells despite the overall Se concentration still lower than the ZIP8-WT and ZIP8-C113S (Figure 3C). Nevertheless, the Se uptake ability of ZIP8 in the S335T mutant remains negligible in the moderate Se exposure setting (Figure 3C).

To examine whether the abilities of ZIP8 mutants to transport Se are associated with their ZIP8 protein expression, we performed immunoblot analysis to detect ZIP8 protein expression in these mutants. Results indicated that when compared with the ZIP8-WT, most of the mutants (ZIP8-C113S, ZIP8-G204C, and ZIP8-S335T) showed equal or stronger ZIP8 expressions where only ZIP8-G38R showed substantially weaker expression of ZIP8 (Figure 3D,E). Therefore, it appears that there is no direct connection between the ZIP8 protein expression level and Se transport ability.

### 2.4. The Role of ZIP8 and Se in Cd Cytotoxicity

It has been previously recognized by researchers that Se can effectively counteract the cytotoxic effect of Cd [31]. Since we have identified ZIP8 as a transporter of both Se and Cd, we next sought to investigate whether ZIP8 and/or Se play a role in counteracting Cd cytotoxicity. As expected, when exposed to CdCl_2_ (2 μM for 12 h), HeLa ZIP8-KO cells transfected with pcDNA3.1-ZIP8-WT showed a remarkable increment (875.4-fold) of intracellular Cd as compared with the cells transfected with pcDNA3.1 empty vector (only up by 205-fold) (Figure 4A). Addition of 4 μM Na_2_SeO_3_ differentially reduced the Cd concentrations in both ZIP8-WT (by 73.6%) and empty vector control cells (by 44.5%) (Figure 4A). Consistent with these findings, cell viability results in Figure 4B indicated that the cell toxicity mediated by Cd was suppressed in the presence of Se, in particular, the suppressive effect of Cd-induced cell cytotoxicity can be found more obviously in the cells expressing ZIP8 protein (Appendix A). To conclude, these results demonstrate that ZIP8 has an essential regulatory effect on cellular Cd and Se uptake that is associated with Cd-induced cell toxicity.

We then used ICP-MS to further investigate the Cd transport ability of the four selected ZIP8 mutations (G38R, C113S, G204C, and S335T), and at the same time, we also assessed whether Se could suppress Cd cytotoxicity in the cells with these ZIP8 mutations. Overall, the results indicated that when treated with 2 μM CdCl_2_ for 12 h, the Cd concentrations were increased by 757 to 1053-fold for ZIP8-WT, -G38R, -C113S, and -G204C cells and only increased by 402-fold for ZIP8-S335T cells (Figure 4C). When treated the cells together with Cd and Se, a significant reduction of Cd levels was observed in the ZIP8-WT, -G38R, -C113S, and -G204C cells but not in the ZIP8-S335T cells (Figure 4D). Surprisingly, we discovered that higher Se levels were detected in all the cells treated with Se and Cd as compared with treated with Se alone (Figure 4E), and among them, the ZIP8-S335T cells which previously showed poor to no Se and Cd transport ability has the highest intracellular Se concentration (Figure 4E,F).

### 2.5. Investigating the Potential Anticancer Effects of Se and Synergistic Anticancer Effect of Se and Cisplatin in Cancer Therapy

The potential of Se in cancer prevention and therapy has been extensively discussed. Although it remains controversial, some studies have reported that Se is an effective anticancer agent [3,32]. In our study, we found that even though 4 μM of Na_2_SeO_3_ showed no cytotoxic effect on the HeLa cells when treated for 12 h (Figure 4B), the same concentration of Se would kill about 75.72% of the cells at 48 h (Figure 5A). The dramatic drop of cell viability at 3 to 4 μM of Se treatment indicated that the anticancer property of Se is not only highly dependent on treatment duration but the concentration of Se is also very critical.

Research has shown that Se can potentiate the anticancer effects of some chemotherapeutic agents, including the first-line chemotherapeutic drug cisplatin [15]. Herein, we further explore the potential of Se on cancer therapy, and evaluate whether Se in combination with cisplatin shows a synergistic effect on HeLa cells. Explicitly, cell viability results obtained from NBB staining assay demonstrated that 4 μM Na_2_SeO_3_ was more effective in killing HeLa cells than 5 μM cisplatin (concentration was selected based on IC50 determined previously) and no synergistic effect was observed between Se and cisplatin (Figure 5B and Appendix A). On the contrary, the HeLa cells survived better in the combined treatment of Na_2_SeO_3_ (4 μM) and cisplatin (5 μM) than the Na_2_SeO_3_ (4 μM) alone treatment (Figure 5B). This unexpected finding implies that cisplatin may alleviate the Se-induced cell toxicity in the HeLa cells. ICP-MS was then performed to see if cisplatin affects Se transport ability in the HeLa cells, and it was revealed that cisplatin treatment resulted in a reduced intracellular Se uptake (Figure 5C).

Another two human lung cancer cell lines (H1299 and H1975) were also tested to verify whether the anticancer effect of Se and cisplatin are cell type-dependent. Collectively, H1299 and H1975 cells did not respond to Se and cisplatin the same way as the HeLa cells: treatment of 4 μM Na_2_SeO_3_ for 48 h was toxic for the HeLa cells but not so much for H1299 and H1975 cells (Figure 5D,E). Again, no synergistic effect between Se and cisplatin was observed in H1299 and H1975 cells (Figure 5D,E).

### 2.6. Clinical Database Analysis: Gene Expressions of ZIP8 and Selenoproteins in Cancers

We next evaluate how relevant is ZIP8 in cancers by utilizing TCGA database [24]. We analyzed the mRNA expression of ZIP8 in 40 types of cancers, and found that for those that showed significant differences in their ZIP8 expressions between tumor and healthy tissues, almost all were up-regulated and only five cancer types (acute myeloid leukemia, kidney renal clear cell carcinoma, liver hepatocellular carcinoma, thymoma, and uveal melanoma) contained a small percentage of tumor samples with down-regulated ZIP8 expression (Figure 6). To further verify that up-regulated ZIP8 expression is a common feature of cancer cells, we detected the protein expression of ZIP8 in normal lung epithelial BEAS-2B cells along with several other cancer cell lines (A549, H1299, H1975, H358, H460, and HeLa), and found the ZIP8 expressions in all these cancer cell lines were indeed higher than the BEAS-2B cells (Appendix A). Given the fact that ZIP8 is involved in cellular Se homeostasis and that Se exhibits antitumor properties, the above findings suggest that the expression of ZIP8 in cancer tissues could be an important predictive factor for Se-based anticancer therapy.

The biological function of Se, as an essential micronutrient, is principally regulated by the synthesis of selenoproteins [3,33]. So far, 25 genes coding for selenocysteine-containing proteins have been identified in humans, and some of these proteins are involved in tumorigenesis [34,35]. To evaluate the potential connections between ZIP8 expression and the selenocysteine-containing proteins in cancers, we obtained and summarized the co-expression data of ZIP8 and selenoprotein genes in a range of cancer types from TCGA database (Figure 7). Overall, it is demonstrated that two selenoproteins, SELENOF (SelF) and SELENOP (SelP), are only positively correlated with ZIP8 across multiple cancer types (23/40 and 19/40, respectively). On the other hand, SELENOV (SelV) is identified as the selenoprotein that only negatively correlated with ZIP8 in 10 out of 40 cancer types. Interesting results can also be found if we focus on individual cancer types. For example, the majority of selenoproteins (18/25) in the upper tract urothelial carcinoma show significant correlations with ZIP8 and these correlations are all positive. Furthermore, little or no correlation between ZIP8 and selenoprotein is observed in certain cancers such as kidney renal papillary cell carcinoma and adrenocortical carcinoma. Taken together, the above data provide new insights into molecular and clinical aspects revolving around ZIP8, Se, selenoproteins, and cancer. Nevertheless, substantial studies are required in the future to assess if Se-based therapy could be more effective in certain cancer types with altered ZIP8 expressions.

## 3. Discussion

ZIP8 was recently reported as a Se transport protein. McDermott et al. [17] used multiple model systems to study the Se transport ability of ZIP8, and they showed that Se uptake was closely associated with ZIP8 expression in all the models tested, including cell cultures and transgenic mice. However, unlike McDermott et al. [17] where the ZIP8 expression of arsenic-transformed human bronchial epithelial BEAS-2B cells was knockdown by shRNA, in the current study, we used CRISPR/Cas9-based KO system to eliminate the endogenous expression of ZIP8 in the HeLa cells. This ZIP8-KO cell model allows us to more explicitly study the functions of ZIP8 and ZIP8 mutations because no interference will occur between the endogenous ZIP8 and ectopic ZIP8-WT or ZIP-mutation. Using the established ZIP8-KO and ZIP8-overexpressed human cell models, we confirmed and verified that ZIP8 is involved in the intracellular transportation of Se.

It is not uncommon that a single-point mutation on a single gene can cause the protein to function abnormally, leading to human diseases [27]. Over the years, a number of human disease-related ZIP8 mutations have been identified. These mutations include V33M (c. 97G>A), G38R (c. 112G>C), C113S (c. 338G>C), G204C (c. 610G>T), S335T (c. 1004G>C), I340N (c. 1019T>A), and A391T (c. 1171G>A) [21,23,36], and cells with ZIP8 containing these mutations appear to have dysregulated ion homeostasis (especially Mn) which can lead to diseases such as type II congenital disorder of glycosylation [37], Leigh syndrome [37], cardiovascular diseases [38], severe idiopathic scoliosis [39], and schizophrenia [40]. Here, for the first time, we discovered that single-point mutations in the ZIP8 gene could differentially affect the uptake of Se: ZIP8 variants G38R, G204C, and S335T showed reduced Se uptake ability whereas variant C113S had a Se transport ability comparable to the ZIP8-WT. These results indicate that ZIP8 is participating in Se homeostasis and that certain mutations in the ZIP8 could disrupt intracellular Se levels, and consequently, may lead to diseases related to Se deficiency or selenosis (Se overdose).

The molecular and cellular mechanisms underlying how mutations in ZIP8 affect the transportation of Se and other essential and nonessential metals remain largely unknown. Studies have shown that some of the nonsynonymous mutations will cause the protein structure of ZIP8 to change and alter the metal binding affinity. For example, it was shown that mutation A391T induced a structural change of the ZIP8 protein and resulted in a less intracellular Cd uptake [38]. Moreover, a report has indicated that disease-associated human ZIP8 mutants V33M, G38R, C113S, G204C, S335T, and I340N are unable to localize to the plasma membrane of the cells. Therefore, the mislocalization of ZIP8 with nonsynonymous mutations may be one of the reasons that some of the ZIP8-mutant cells in this study showed dysregulated Se homeostasis, but further ZIP8 localization test has to be carried out to check this possibility.

Results obtained from the ICP-MS (Figure 2A–C and Figure 3B,C) and immunoblot analysis (Figure 3D,E) indicated the metal transport abilities of the ZIP8-WT and ZIP8-mutant cells are not entirely correlated with the ZIP8 protein expressions. This is partly because the ZIP8 protein is not the sole transporter for the metals tested—the nutrient and metal transportation networks in the cells are extremely complicated and virtually each of the elements can be transported by more than one transporter [41]. Also, we previously chose the first 100 amino acids of ZIP8 N-terminal region as the immunogen for antibody production, and therefore, we cannot exclude the possibility that some mutations (e.g., the G38R mutant) may affect the affinity of ZIP8 antibody. Nevertheless, the critical role of human ZIP8 in transporting Se has been clearly demonstrated in the current study.

We next explored the potential of Se in disease prevention (Cd cytotoxicity) and therapy (cancer), from a ZIP8 point of view. Here, we found that Se could suppress Cd-induced cytotoxicity by reducing the entry of Cd via ZIP8 (Figure 4A–C). These results generally coincide with recent studies which reported that Se mitigated Cd-induced toxicity by regulating selenoprotein synthesis through promoting selenoprotein transcription [31,42]. In addition, an interesting discovery has been observed: Cd could promote intracellular Se uptake (Figure 4F,G). The above findings warrant further epidemiological and clinical studies to assess whether Se-deficient people are more prone to Cd-induced diseases and if Cd exposure is linked to selenosis. Also, since Cd is a widespread environmental pollutant and carcinogen that poses a serious threat to human health, especially in smokers and people working or living in Cd-polluted environments [43], research on the mechanistic association between ZIP8, Se, and Cd is urgently needed.

Se and Se-based compounds are considered promising candidates in chemoprevention and cancer therapy. Some of the Se compounds exhibit not only excellent anticancer properties but also remarkable tumor specificity [44]. The anticancer abilities of Se are mainly achieved by the direct or indirect antioxidant properties of Se [45]. Studies have indicated that Se may lower the adverse effects and increase treatment efficacy when it is used in combination with other anticancer agents [3]. Specifically, it has been reported that Se could potentiate the efficacy of cisplatin, a first-line chemotherapeutic drug for many cancer types [8,32]. However, despite the benefits of Se in chemoprevention and cancer therapy mentioned above, the utility and safety of Se and Se compounds for chemoprevention and cancer treatment remains uncertain due to the two-sided effect on cancer cells: on the one hand, Se shows cancer cell inhibitory properties, but on the other hand, Se may also promote cancer cell growth [9,46]. These contradictory results may be due in part to different chemical forms and bioavailability of Se, cancer cell types, and experimental conditions. Here, we showed that the anticancer effects of Se could be higher than the chemotherapeutic drug cisplatin in selected cancer cell lines and that the efficacy of Se is strictly concentration-dependent (Figure 5). Also, we did not observe a synergistic effect of Se and cisplatin when both were used together on the HeLa, H1299, and H1975 cell lines. However, this could be due to the limited range of Se concentrations tested in our study. Therefore, for Se-based compounds to be clinically used in cancer therapy, systematic and comprehensive screenings are required to obtain the optimal treatment conditions of Se against cancer cells with various phenotypic and genotypic backgrounds. Additionally, since ZIP8 is involved in the uptake of Se, further studies should be also considered for the potential involvement of ZIP8 in Se-based anticancer therapy.

Our study also provides clinical insights on how ZIP8 may be involved in Se-based cancer therapy. In Figure 6 and Figure 7, we analyzed the gene expression of ZIP8 and 25 selenocysteine-containing proteins in clinical samples across multiple cancer types. We found that cancer tissues in almost all cancer types are inclined to have up-regulated ZIP8 expressions, and this may have implications for precision medicine since Se may more precisely target the tumor cells instead of the healthy cells. Selenium nanoparticles (SeNPs) possess a strong pro-oxidant property, and hyper-accumulation of SeNPs can generate potent therapeutic effects for cancer [47]. In addition, we identified selenoproteins that are positively or negatively co-expressed with ZIP8 in multiple cancer types. The roles of selenoproteins in cancer have been reported [36,48]. Overall, data summarized in Figure 7 provide guidance for future research by showing which selenoproteins in a specific cancer type are worthy of further investigation.

## 4. Materials and Methods

### 4.1. Cell Lines and Culture Conditions

Human cervical cancer (HeLa) and lung cancer (H1299 and H1975) cell lines were purchased from the American Type Culture Collection (ATCC) (Rockville, MD, USA). All cells were routinely grown in MEM or RPMI medium containing 10% fetal bovine serum (FBS) and 1% penicillin/streptomycin at 37 °C in a 5% CO_2_ incubator as recommended by ATCC. For experiments containing additional Se, 2.5 mM NaHCO_3_ and 150 μM ZnCl_2_ were added into the culture medium.

### 4.2. Generation of ZIP8-KO HeLa Cell Model Using CRISPR/Cas9 Genome Editing Technology

HeLa cells were transfected using the pSpCas9(BB)-2A-Puro (PX459) plasmid plus the sgRNA sequence. Primers of sgRNA are listed in Appendix A. The cells were digested and selected by puromycin (1 μg/mL, Sigma-Aldrich, Taufkirchen, Germany) after transfection for 48 h, and refreshed with MEM medium containing FBS and puromycin every day. After several passages, clones were picked, seeded into a 12-well plate (one clone each), and cultured. The efficiency of KO was detected using immunoblot analysis and PCR. Genomic DNA was extracted with Thermo Genomic DNA Kit (Thermo Fisher Scientific, Waltham, MA, USA) according to the manufacturer’s instructions, and gene KO condition was verified by touchdown PCR using Premix Taq Hot Start on a PCR. The PCR conditions were 94 °C for 3 min, a touchdown step of 15 cycles at 94 °C for 30 s, 65 °C for 30 s, and 72 °C for 1 min with a 1 °C decrease every cycle in annealing temperature, followed by 25 cycles at 94 °C for 30 s, 50 °C for 30 s, and 72 °C for 1 min, and a final extension at 72 °C for 10 min. The PCR products were examined by 1% agarose gel electrophoresis with gel-red staining, and visualized by a UV illuminator; subsequently, the PCR products were sent for sequencing to further confirm the homozygous KO of the ZIP8 gene.

### 4.3. Plasmids and Cell Transfection

pSpCas9(BB)-2A-Puro (PX459) plasmid, obtained from Addgene, was used to KO ZIP8. Single guide RNAs (sgRNAs), listed in Appendix A, were designed using an online web-based tool (http://crispor.tefor.net/). To overexpress the ZIP8, the coding sequence of human ZIP8 was amplified by PCR using cDNA of normal human bronchial epithelial BEAS-2B cells as a template, and inserted into pcDNA3.1 vector through BamHI and EcoRI. Its mutations were constructed following the molecular cloning guidelines and using the QuickChange site-directed mutagenesis kit (Agilent#210519, Palo Alto, Santa Clara, CA, USA) according to manufacturer’s instructions. The primers used are shown in Appendix A. The verification of all plasmids was subjected to sequencing. Plasmid transfection was carried out in HeLa ZIP8-KO cells following the manufacturer’s instructions.

### 4.4. Cytotoxicity Assay

Cell viability was performed by the NBB staining assay. Briefly, 100 μL cell suspension containing 1 × 10^6^ cells/mL was added in each well of a 96-well plate, and cells were cultured in an incubator for 24 h. Cells were then exposed to drugs of interest for a duration as indicated prior to NBB staining assay. During NBB assay, cells were fixed with 4% formaldehyde solution (10% formalin) for 8 min, followed by staining with 0.05% NBB solution for 25 min at room temperature. Then, cells were washed with distilled water thrice, and finally, 50 μL of 50 mM NaOH was added. The absorbance of the cell suspension was measured at 595 nm using a 96-well multiscanner (Thermo Scientific Multiskan FC, Thermo Scientific, Waltham, MA, USA).

### 4.5. Immunoblot Analysis

Cells were lysed with NP-40 lysis buffer containing a protease inhibitor cocktail and sonicated for 6 s each for three times. The cell lysates were then centrifuged at 16900× *g* for 10 min at 4 °C. Then, 30 μg protein were quantified and used for immunoblot analysis as described previously [49]. The following antibodies were used: anti-ZIP8 (1:500; produced by our lab [50]), anti-β-actin (1:10,000; #A5441, Sigma-Aldrich, Taufkirchen, Germany), anti-rabbit IgG-HRP (1:10,000; #sc-2004, Santa Cruz Biotechnology, Santa Cruz, CA, USA), and anti-mouse IgG-HRP (1:10,000; #sc-2005, Santa Cruz Biotechnology, Santa Cruz, CA, USA).

### 4.6. Inductively Coupled Plasma Mass Spectrometry (ICP-MS)

The quantifications of intracellular Se, Mn, Zn, and Cd levels were performed using Agilent 7900 ICP-MS (Agilent Technologies Inc., Santa Clara, CA, USA) using argon as a plasma gas. Briefly, the cells to be measured were digested by trypsin, washed with 1 × PBS thrice and lysed in Milli-Q H_2_O, followed by quantification of cell lysate. 700 μg of cell lysates were taken out and subsequently added into a polytetrafluoroethylene tube for air drying at 60 °C in an oven overnight. In order to decompose the organic matrix, samples were processed in 100 μL 65% HNO_3_ and 100 μL H_2_O_2_ at 80 °C for nitrification, then cooled to room temperature. Then, all samples were diluted in 2% HNO_3_ to a final volume of 5 mL and eventually quantification of intracellular elements by ICP-MS. To validate the method, we evaluated the linearity, limit of detection (LOD), and the limit of quantification (LOQ) [51]. When cell samples were assessed by ICP-MS, a blank and a trace elements serum L-2 RUO were measured for each time as quality control.

At least seven calibration points were used for external calibration of each of the elements, all elemental concentrations of samples have been considered. The multi-element calibration standard solution (10 μg/mL) was diluted by 2% HNO_3_ solution to prepare the working standards at multiple concentrations of 1, 5, 10, 50, 100, and 500 μg/L, which would be re-prepared for each time before measuring the samples. The internal standards were applied to correct signal drift. The multi-element solution (100 μg/mL) was diluted with 2% HNO_3_ solution to prepare the internal standard solutions, germanium, indium, and scandium were used as the internal isotopes for quantification. In addition, the LOQ and the LOD of each element were determined by measuring elemental concentrations until detecting digestion blanks for each time (Appendix A). Correlation coefficients of all elemental calibration curves were above 0.999.

### 4.7. Immunofluorescence Microscopy

Cells were seeded on coverslips in a 6-well plate, fixed with 4% paraformaldehyde for 15 min at room temperature, and rinsed with phosphate-buffered saline (PBS) containing Ca^2+^ and Mg^2+^ thrice. Then, cells were permeabilized with pre-cooled 0.02% Triton X-100 for 10 min on ice and again rinsed with PBS containing Ca^2+^ and Mg^2+^ thrice. Next, cells were blocked with 5% bovine serum albumin (BSA) in PBS at room temperature for 2 h. After blockage with 5% BSA, cells were incubated in the dark with ZIP8 (1:200) and DMT1 (1:150, #166884, Santa Cruz Biotechnology, Santa Cruz, CA, USA) primary antibodies for more than 8 h at 4 °C. Subsequently, cells were incubated with secondary antibodies (Alexa Fluor secondary 488 and 595; Invitrogen) for 2 h at 1:800 dilution in the dark; The nuclei were stained with Hoechst 33258 (#B1155, Sigma–Aldrich, Taufkirchen, Germany) for 15 min. Immunofluorescence images were taken by a LSM800 confocal laser scanning microscope (Zeiss, Oberkochen, Germany) at 400× magnification.

### 4.8. Clinical-Based Bioinformatics Analysis

mRNA expression data of ZIP8 and 25 genes coding for selenocysteine-containing proteins in patients from 40 cancer types were obtained from TCGA database. The mRNA expression z-score threshold was set at ±2.0. For co-expression analysis, only data with *p* ≤ 0.05 and spearman’s correlation > 0 (positive correlation between ZIP8 and selenoprotein) or <0 (negative correlation between ZIP8 and selenoprotein) were considered significant.

### 4.9. Statistical Analysis

Statistical analysis was performed using the GraphPad Prism^®^ 7 software (v6.02, GraphPad Software Inc., San Diego, CA, USA). All quantitative data were expressed as means ± standard deviation of at least three independent experiments unless mentioned otherwise. Statistical significance between groups was analyzed using a two-tailed Student’s *t-test*, one-way ANOVA, or two-way ANOVA. A *p* ≤ 0.05 was used as the criterion for statistical significance.

## Figures and Tables

**Figure 1 ijms-22-11399-f001:**
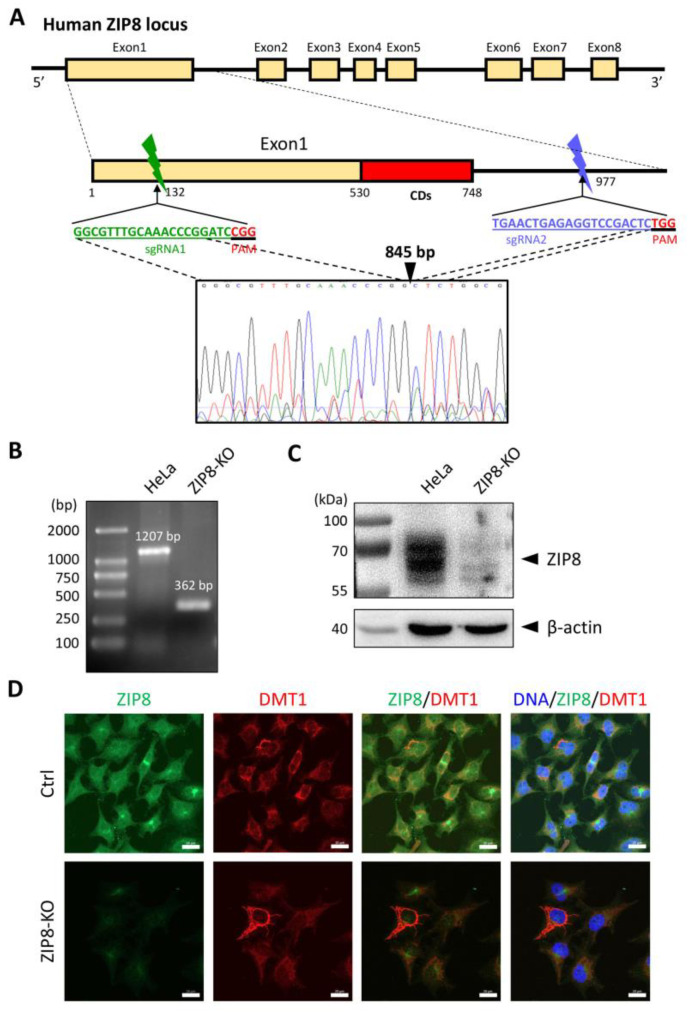
Generation of a ZIP8-KO HeLa cell line using the CRISPR/Cas9 genome editing technology. (**A**) Schematic of the CRISPR/Cas9 system to KO ZIP8. Two sgRNAs (green and blue bars) were designed to pair with the targeted DNA in order to delete a desire region in the human ZIP8 locus (part of exon1 and intron1). Each of the sgRNAs contains 20 bp single guide sequence followed by an essential PAM sequence (5′-NGG adjacent motif), and the site of double-stranded breaks induced by Cas9 is located about 3 nt upstream of the PAM sequence. DNA sequencing chromatogram shows that 845 bp of the ZIP8 gene has been deleted and the DNA junctions are connected via non-homologous end-joining (filled triangle). (**B**) Genomic DNA of HeLa and ZIP8-KO cells were extracted and amplified by PCR. (**C**) ZIP8 protein levels in HeLa and ZIP8-KO cells were analyzed by immunoblot analysis. Total cell lysates were resolved by SDS-PAGE, transferred onto polyvinylidene difluoride membrane, and incubated with antibody against ZIP8. β-actin was used as the indicated loading control. (**D**) The localization of ZIP8 was verified by immunofluorescence analysis in HeLa parental and HeLa ZIP8-KO cells using ZIP8 and DMT1 antibodies. DMT1 is a marker located in the membrane. The images were captured by confocal fluorescence microscopy. The merged images of ZIP8 (green) and DMT1 (red) reflect the colocalization area (yellow), and the blue color represents the nuclei stained with Hoechst 33258. Scale bar = 20 µm.

**Figure 2 ijms-22-11399-f002:**
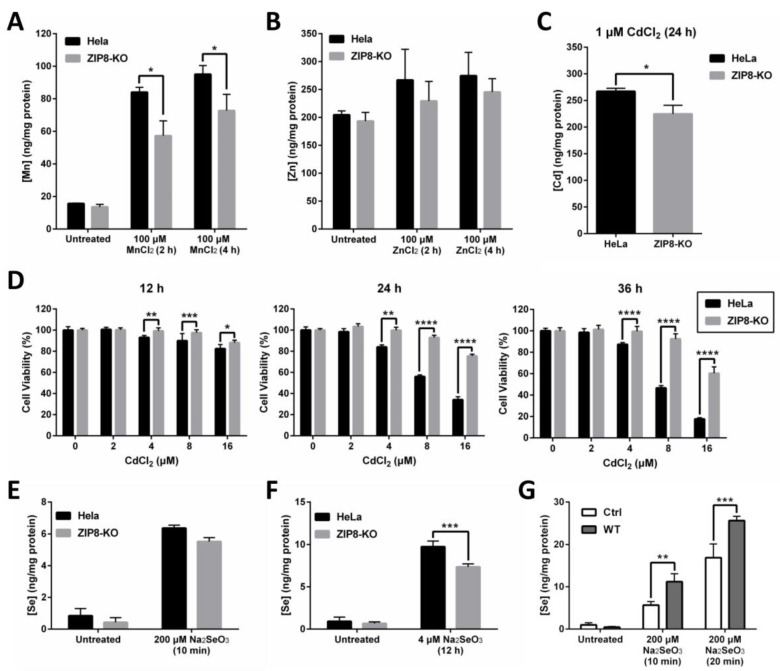
ZIP8-KO cells show decreased ability in intracellular Mn, Zn, Cd, and Se uptakes. (**A**–**C**) Intracellular uptake of Mn (**A**), Zn (**B**), or Cd (**C**) in HeLa parental and ZIP8-KO cells. After treating the cells with 100 μM MnCl_2_ or ZnCl_2_ for 2 or 4 h or 1 μM Cd for 24 h, the intracellular content of Mn, Zn, or Cd was assessed by ICP-MS. (**D**) The HeLa parental and ZIP8-KO cells were exposed to increasing amounts of CdCl_2_ (0, 2, 4, 8, or 16 μM) for 12, 24, and 36 h. NBB staining assay was performed to determine the cell viability. Data are representative of three experiments, and error bars represent standard deviation of six replicates. (**E**,**F**) HeLa parental and ZIP8-KO cells were treated with 200 μM Na_2_SeO_3_ for 10 min (**E**) or 4 μM Na_2_SeO_3_ for 12 h (**F**), followed by measurement of Se cellular contents using ICP-MS. (**G**) HeLa ZIP8-KO cells were transfected with pcDNA3.1 empty vector (as a control) or pcDNA3.1-ZIP8-WT. Cells were exposed to 200 μM Na_2_SeO_3_ for 10 or 20 min, and then the intracellular Se concentration was measured by ICP-MS. Data are representative of two experiments, and error bars represent standard deviation of three replicates. *p*-value less than 0.05 was considered to be statistically significant. * *p* < 0.05, ** *p* < 0.01, *** *p* < 0.001, **** *p* < 0.0001.

**Figure 3 ijms-22-11399-f003:**
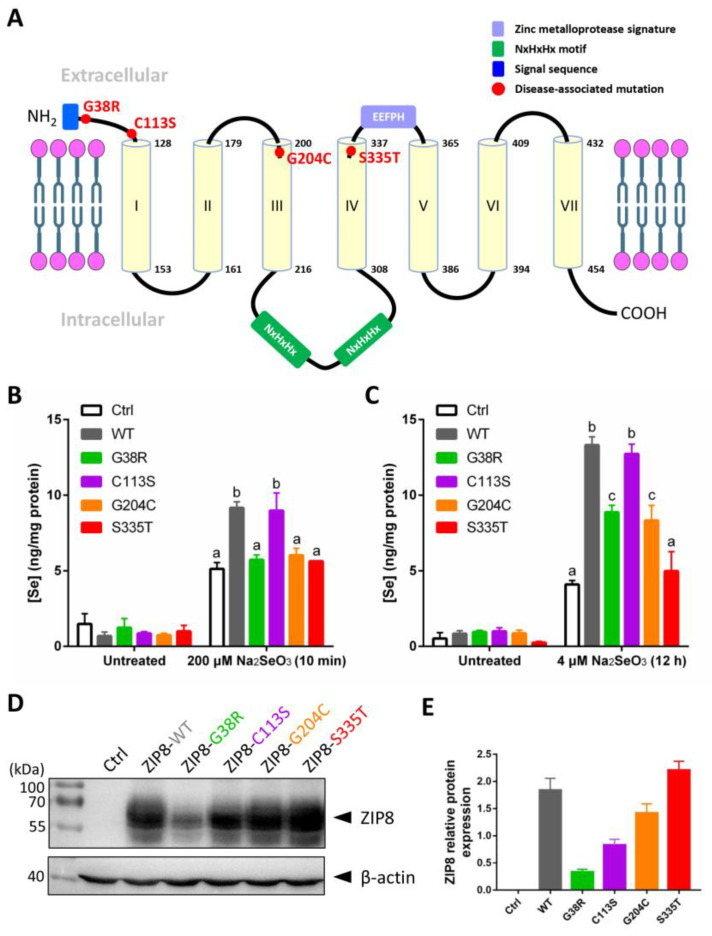
Se transport ability and protein expression level of ZIP8 variants. (**A**) Structural diagram of ZIP8 protein based on in silico predictions. It is predicted that the ZIP8 protein has seven transmembrane domains (TMD, yellow cylinder). The sites of four mutations (G38R, C113S, G204C, and S335T) in ZIP8 were labeled in red color. The N-terminus of ZIP8 has a signal sequence containing 22 amino acids (blue frame), and the protein possesses a long intracellular loop between TMD III and TMD IV containing two copied of NxHxHx domains (green frame) responsible for interacting with transition metals such as copper, nickel, and zinc [29,30]. (**B**,**C**) The HeLa ZIP8-KO cells were transient-transfected with pcDNA3.1-ZIP8-WT or ZIP8 mutants (G38R, C113S, G204C, and S335T) for 36 h, and then cells were treated with 200 μM Na_2_SeO_3_ for 10 min (**B**) or 4 μM Na_2_SeO_3_ for 12 h (**C**), followed by detection of intracellular Se concentration using ICP-MS. A significant difference of *p* < 0.05 between any two data columns marked with different alphabets. (**D**) The HeLa ZIP8-KO cells were transient-transfected with pcDNA3.1 (Ctrl) or pcDNA3.1-ZIP8-WT or mutants (G38R, C113S, G204C, and S335T) for 24 h, and proteins from these cells were harvested for immunoblot analysis to detect the protein expression level of ZIP8. β-actin was used as the loading control. (**E**) The quantifications of ZIP8-WT and other mutants (G38R, C113S, G204C, and S335T) in immunoblot result.

**Figure 4 ijms-22-11399-f004:**
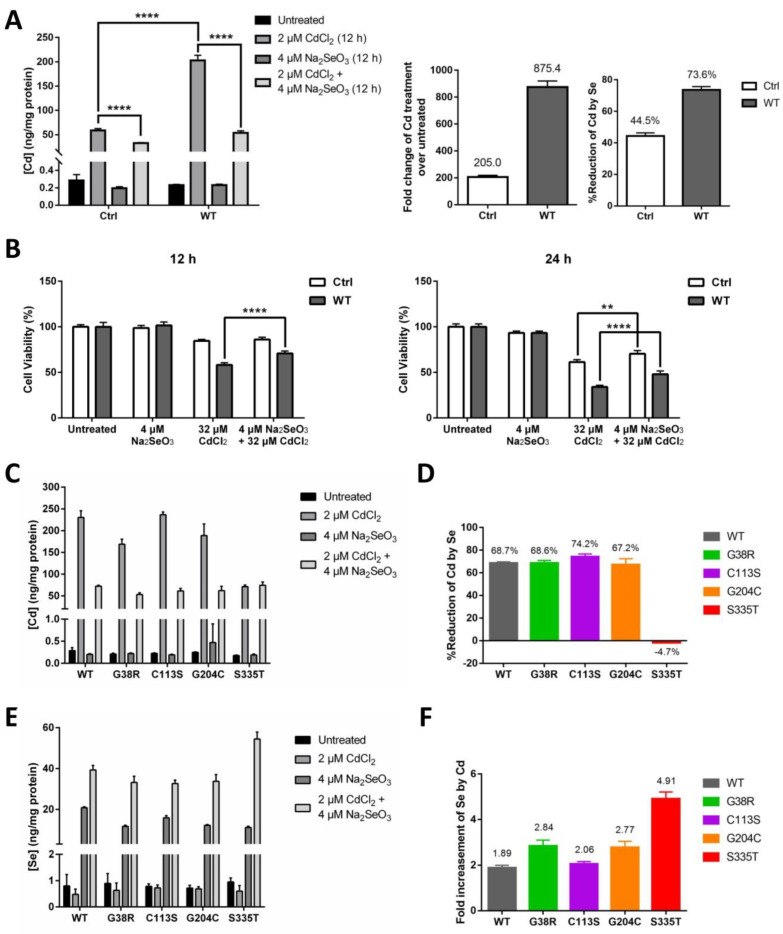
Se reduces Cd-induced cytotoxicity via decreasing the level of intracellular Cd transported by ZIP8. (**A**,**B**) The HeLa ZIP8-KO cells were transfected with a control plasmid (pcDNA3.1 empty vector) or pcDNA3.1-ZIP8-WT overexpression plasmid for 36 h, and cells were treated with various concentrations of CdCl_2_ and/or Na_2_SeO_3_ for 12 h. Intracellular Cd content was measured by the ICP-MS (**A**), and cell viability was determined by the NBB staining assay (**B**). (**C**,**E**) Intracellular Cd (**C**) or Se (**E**) concentration was determined by ICP-MS after exposed to 2 μM CdCl_2_ and/or 4 μM Na_2_SeO_3_ for 12 h in HeLa ZIP8-KO cells overexpressing ZIP8-WT or ZIP8-mutant. (**D**) Percentage reduction of intracellular Cd level between the Cd and Cd + Se treatment groups from (**C**). (**F**) Fold change of intracellular Se level between the Se and Cd + Se treatment groups from (**E**). Data in (**B**) are representative of three experiments, and the error bars represent standard deviation of six replicates. Two-way ANOVA (A, B) and one-way ANOVA (**A**) were performed, and a *p*-value less than 0.05 was considered to be statistically significant. ** *p* < 0.01, **** *p* < 0.0001.

**Figure 5 ijms-22-11399-f005:**
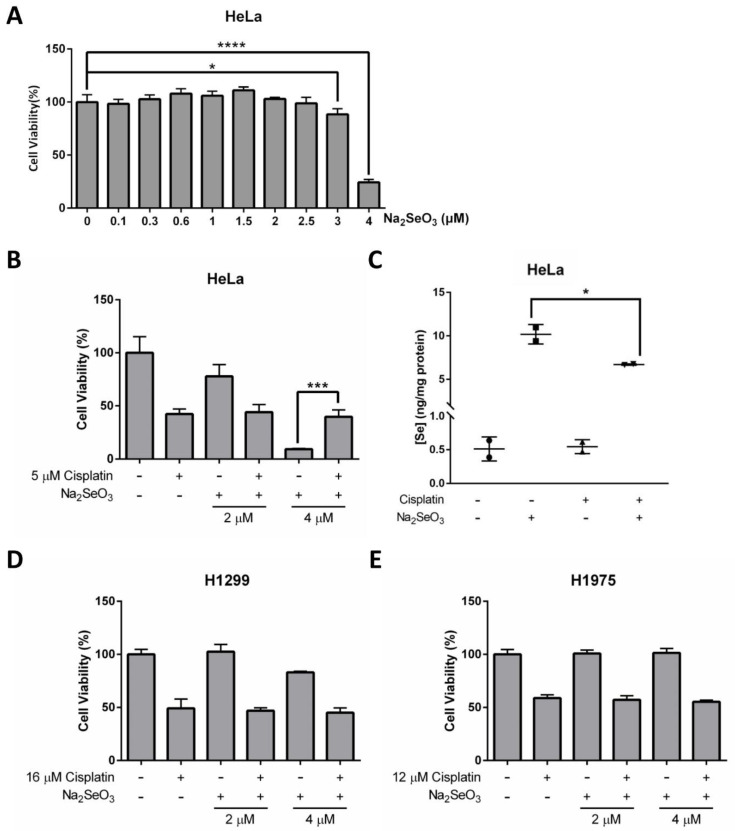
Anticancer effects of Se and cisplatin on HeLa, H1299, and H1975 cells. (**A**) Cell viability of HeLa cells treated with increasing doses of Na_2_SeO_3_ for 48 h. (**B**) Cell viability of HeLa cells incubated with 5 μM cisplatin and/or different doses of Na_2_SeO_3_ (2–4 μM) for 48 h. (**C**) Se concentration in HeLa cells treated with 5 μM cisplatin and/or 4 μM Na_2_SeO_3_. (**D**,**E**) Cell viability of H1299 cells (**D**) and H1975 cells (**E**) incubated with 16 μM (D) or 12 μM (E) cisplatin and/or different doses of Na_2_SeO_3_ (2–4 μM) for 48 h. Cell viability was measured by NBB staining assay (**A**,**B**,**D**,**E**), and Se concentration was measured by ICP-MS (**C**). One-way ANOVA (**A**–**C**) was performed, and a *p*-value less than 0.05 was considered to be statistically significant. * *p* < 0.05, *** *p* < 0.001, **** *p* < 0.0001.

**Figure 6 ijms-22-11399-f006:**
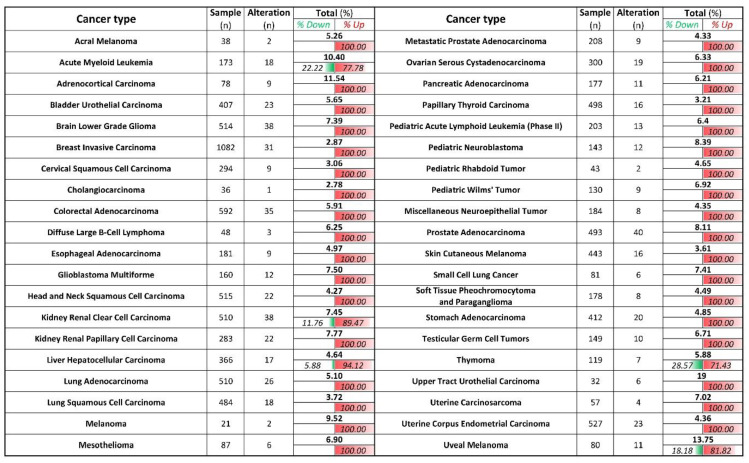
ZIP8 is up-regulated in cancer tissues and cell lines. ZIP8 mRNA expressions in 40 types of cancer were obtained from TCGA database. z-score threshold was set at ±2.0. The number of % alteration in each of the analyzed datasets was listed and further divided into the ratio of down-regulated and or up-regulated expression.

**Figure 7 ijms-22-11399-f007:**
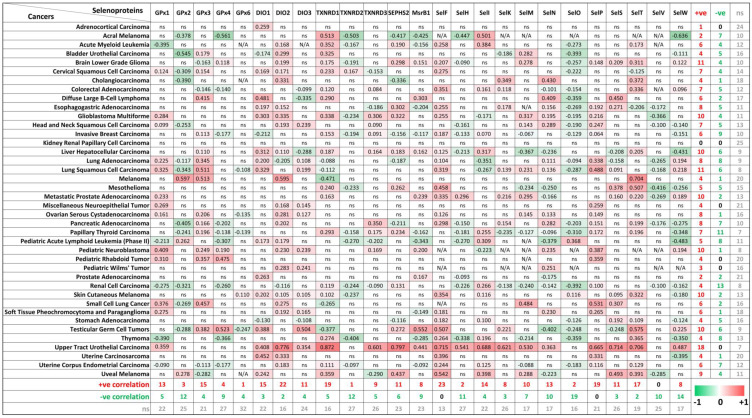
Co-expressions of ZIP8 and selenoprotein genes in cancers analyzed by spearman’s correlation. Co-expression data of ZIP8 and 25 selenocysteine-containing protein genes in 40 types of cancers were obtained from TCGA database. A spearman’s correlation value greater than 0 indicates a positive correlation (red) and value less than 0 indicates a negative correlation (green). Only values with *p*-value less than 0.05 were shown. ns: non-significant; N/A: not available.

**Table 1 ijms-22-11399-t001:** Information of selected disease-associated ZIP8 single-point mutations examined in this study.

Site(aa)	Type	Variant	Disease	References
38	Homozygous variant	c.112G>C (p. Gly38Arg)	Type II congenital disorder of glycosylation	[21,23,28]
113	Homozygous variant	c.338G>C (p. Cys113Ser)	Leigh syndrome	[23]
204	Heterozygous variant	c.610G>T (p. Gly204Cys)	Type II congenital disorder of glycosylation	[22,23]
335	Heterozygous variant	c.1004G>C (p. Ser335Thr)	Type II congenital disorder of glycosylation	[22,23]

## Data Availability

All data generated or analyzed during this study are included in the main manuscript and Appendix A.

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
