# Peer review of "The Impact of ZIP8 Disease-Associated Variants G38R, C113S, G204C, and S335T on Selenium and Cadmium Accumulations: The First Characterization"

_ijms, 2021, doi:10.3390/ijms222111399_

Round 1

Reviewer 1 Report

This article aims at exploring the ability of ZIP8 as a modulator of cellular Se levels and at providing insights on how ZIP8 and Se may involve in disease prevention and therapy. The manuscript is overall well written and the experimental design well conducted. Although being recommended for publication, the manuscript has few points which the authors should take into consideration:

Keywords:

Instead of using SLC39A8, ZIP8 should be selected as a keyword. With the exception of the abstract, throughout the manuscript the SLC39A8 nomenclature was not used.

Results:

  • The results proving the ZIP8 gene of HeLa cells had no mutations (lines 141-143) should be part of the supplementary material;
  • Figure2C: why the levels of Cd were only measured after 24h whereas uptake levels of Mn and Zn were measured at 2h and 4h? Moreover, authors should clarify the difference timepoints used among the metal ions uptake (e.g. 2h/4h vs 24h);
  • Figure 4A: Statistical significance should be added to the graphical representation to better followed the results explained on 2.4. section;
  • Figure 4A/4D/4F: Coherency should be given to graphical representation of similar outputs (e.g. the values above the bars when showing %reduction and fold increase/change). Why those figures do not have error bars?;
  • To better understanding of the reader, in line 257 it should be referred Figure 4B after stating that no cytotoxic effect was found when HeLa cells were incubated with 4uM of Na2SeO3 after 12h;
  • Authors claim that the dramatic increase of cytotoxicity at 3 to 4uM indicated that Se anticancer activity is highly dependent of treatment duration and concentration however no data has been shown from intermediate concentrations between 3 and 4uM (lines 258-260);
  • Dose-response curve of cisplatin should be part of supplementary material to support the choice of 5uM as the concentration to be evaluated as combined treatment with Se (lines 266-267; Figure 5B).

Author Response

Please see the attachment,

Reviewer 2 Report

Liang and colleagues studied in vitro mechanism of Se and Cd accumulation via the metal cation symporter ZIP8 (SLC39A8) using 2D cultures. Authors employed the CRISPR/Cas9 system to knockout the ZIP8 gene in HeLa cells to study the accumulation of Se, Cd in WT vs KO cells and different ZIP8 single-point mutations.

The successful generation of the ZIP8-KO model has been validated with PCR, Westernblot and DNA sequencing. 

Authors first demonstrated that ZIP8-KO cells accumulate fewer quantities of Mn, Zn, Cd and Se than untransformed WT HeLa cells. Furthermore, the authors showed that ZIP8-KO cells are less sensitive to Cd toxicity than WT HeLa cells. When ZIP8-KO cells were transfected with either the ZIP8-WT gene or 4 different ZIP8 mutants, differential increases in intracellular Se and Cd levels were observed after Cd and Se incubation. The clinical relevance for these findings is at this stage unclear, but the present work contributes to the knowledge on ZIP-8 symporter and its mutations. 

  1. The ICP-MS instrumentation in the method section should contain more details – especially, it should be reported if the present analytical procedure was properly validated, or if it is following previously validated method for Se, Mn, Cd, and Zn quantifications in cells. Moreover, authors should report the types and concentrations of the internal standards used for the measurements and calibration curves. To assure the reproducibility of the study, authors should report which isotopes were used for the quantification and what are the limits of quantifications and detection for each studied metal.
  2. How many off-target sites were predicted for sgRNA that was used for ZIP8KO Crispr/Cas9 experiment?
  3. What is the doubling time for HeLa-WT cells and its ZIP8KO variant? Is population doubling time comparable?
  4. Line 459 reports that 1 ml of cell suspension 1 x 10^6 was added in each well of 96-well plate. There may be a mistake in both volume and cell concentration since the standard well of 96-well plate has a maximum volume of approx. 0.4 mL. If a non-standard type microplate was used, please mention it in the methodology section.
  5. The authors analyzed the quantity of metal using 700 ug of cell lysate. Please explain if cells were washed before harvesting, how cells were harvested, and if the weight (700 ug) refers to wet cell mass (after harvesting) or dry cell lysate. 
  6. Authors showed in Figure 2D the concentration- and time-dependent effects of CdCl2 on HeLa-WT and HeLa-ZIP8KO cells. Same experimental data should be generated for Na2SeO3 as well because the authors show only the effects of Na2SeO3 on HeLa-WT cells at 48h (Figure 5A). Figure 5A clearly shows that 4 uM of Na2SO3 is highly toxic for HeLa cells. Authors used this concentration for subsequent experiments because at 12 h did not observe significant changes in cell proliferation, however, the same applies for chemotherapeutics – it is very difficult to observe clear effects of compounds at 12 h in cells with the population doubling time of 24-30 h, which is a case of HeLa cells. Furthermore, it would be reasonable to expect that HeLa-ZIP8KO cells would present higher resistance to Na2SeO3 effects, and when these KO cells are transfected with WT-ZIP8, the cell viability decreases again (but Figure S1 shows almost no differences between HeLa-ZIP8KO vs HeLa-ZIP8KO transfected with pcDNA3.1.-ZIP8-WT). How do you explain these contradictory findings?
  7. Related to point 4, experiments generated for cisplatin and cisplatin + Na2SeO3 show that the addition of cisplatin decreases the toxicity of 4 uM Na2SeO3 in HeLa cells (Figure 5B). These experiments should be performed also in HeLa-ZIP8KO cells because it is also clear that cisplatin decreases the intracellular concentration of Se (Figure 5C) - is this due to ZIP8 or other transporters? 
  8. Authors reported that in S335T mutation, Se addition does not decrease the Cd levels compared to WT and G38R, C113S and G204C mutants, and the addition of the Cd notably increases Se levels in S335T mutant (almost 5-fold). How do you explain these findings? Figure 3D shows that ZIP8-S335T variant may have the highest expression among all other prepared ZIP8-variants. Is this due to inconsistent transfection efficiency or affinity of ZIP8 antibody to different variants? Please provide quantifications of ZIP8 from all western blots.
  9. The results section already discusses the results along with the text. Therefore, I would recommend using the “results and discussion” approach, since the current “discussion” is rather a conclusion of the work.

Some minor points.

  • I would recommend the inclusion of the graphical abstract summarizing your findings.
  • Line 166 contains “cells were stimulated”, however, the term “stimulation” is usually used when studying agonists/antagonists and receptor´s pharmacology. Please use “cells were incubated” or “ cells were exposed to”
  • Line 58 “plenty of attention” and line 270 “this strange finding” use non-scientific and informal language. 
  • It appears that the article uses an incorrect template and page numbering – some pages are on the template of Biomedicines MDPI journal template, possibly from the previous submission.

Round 2

Reviewer 2 Report

Thank you, I have no further comments.

Author Response

Thank you reviewer 2 so much for your time and satisfaction on our revised manuscript.